# A Numerical Investigation of the Geometric Parametrisation of Shock Control Bumps for Transonic Shock Oscillation Control

**Jack A. Geoghegan** *⬨, **Nicholas F. Giannelis** ⬨ and **Gareth A. Vio** ⬨

School of Aerospace, Mechanical & Mechatronic Engineering, The University of Sydney,
Sydney, NSW 2006, Australia; nicholas.giannelis@sydney.edu.au (N.F.G.); gareth.vio@sydney.edu.au (G.A.V.)
* Correspondence: jack.geoghegan@sydney.edu.au

**Abstract:** At transonic flight conditions, shock oscillations on wing surfaces are known to occur and result in degraded aerodynamic performance and handling qualities. This is a purely flow-driven phenomenon, known as transonic buffet, that causes limit cycle oscillations and may present itself within the operational flight envelope. Hence, there is significant research interest in the development of shock control techniques to either stabilise the unsteady flow or raise the boundary onset. This paper explores the efficacy of dynamically activated contour-based shock control bumps within the buffet envelope of the OAT15A aerofoil on transonic flow control numerically through unsteady Reynolds-averaged Navier–Stokes modelling. A parametric evaluation of the geometric variables that define the Hicks–Henne-derived shock control bump will show that bumps of this type lead to a large design space of applicable shapes for buffet suppression. Assessment of the flow field, local to the deployed shock control bump geometries, reveals that control is achieved through a weakening of the rear shock leg, combined with the formation of dual re-circulatory cells within the separated shear-layer. Within this design space, favourable aerodynamic performance can also be achieved. The off-design performance of two optimal shock control bump configurations is explored over the buffet region for $M = 0.73$, where the designs demonstrate the ability to suppress shock oscillations deep into the buffet envelope.

**Keywords:** transonic aerodynamics; transonic shock buffet; flow control; shock control bumps; Reynolds-averaged Navier–Stokes simulation

## 1. Introduction

Transonic shock buffet is a phenomenon where an interaction between shock waves and the separated shear-layer leads to a self-sustained periodic shock motion. In particular, this instability is present on a variety of aerofoil and wing geometries when subjected to flows within a narrow band of transonic free-stream Mach number/incidence angles and has been studied in detail experimentally and numerically since its discovery [1–5]. It has been observed to exist as a purely flow-driven feature, whereby the sonic region expands and contracts along the aerofoil chord, paired with a complimentary fluctuation in the separated shear-layer aft of the normal shock foot, resulting in a shock oscillation equilibrium. The dominant frequency mode in this interaction is that of the shock motion, which occupies the low frequency bandwidth typical of the low order structural natural frequencies present in most commercial aircraft wings, and hence, due to the intense flow-field perturbation, results in large amplitude oscillation in aerodynamic forces and moments. When this effect is considered in an aeroelastic framework, the Shock-Wave/Boundary Layer Interaction (SWBLI) present under transonic flow conditions can give rise to Limit Cycle Oscillations (LCO), which ultimately result in diminished

fatigue life [6,7]. As such, a significant area of research has evolved around the understanding and development of flow control devices to counter shock oscillation and, where possible, improve on total pressure recovery [8–13].

The research into transonic shock oscillation control can typically fall into one of three main categories, defined by the method with which they interact with the local flow-field around the aerofoil. Further, each of these control methodologies can either be passive or active. The main approaches are summarised as follows:

- Up-stream boundary layer energisers such as mechanical or fluidic-injection Vortex Generators (VGs) [10,14], where the objective is to offset the separation of the downstream shear-layer.
- Trailing edge pressure and flow control such as the Trailing Edge Deflector (TED) [15] or trailing edge flap deflections [16,17], which aim to alter the shock dynamics by influencing the separated shear-layer directly.
- Direct shock control through augmentation of the normal shock that forms at the rear of the supersonic domain by introducing a $\lambda$-shock structure. These techniques include introducing a cavity with a porous plenum [9,18], slotted cavities, and more recently, Shock Control Bumps (SCBs)

Of the existing control methodologies, each has shown some promise in either offsetting the buffet onset boundary or directly controlling the shock in given design conditions. However, each may suffer other unintended consequences such as diminished off-design aerodynamic qualities or exaggeration of the buffet phenomenon at the shifted onset. The SCB in particular is promising as it offers better total pressure recovery due to its ability to decelerate the flow more isentropically through the $\lambda$-shock, although its benefits are very sensitive to geometry and position relative to the mean shock [19,20].

With the development of SCBs in the last two decades, a variety of different geometries has emerged and are normally defined as either wedge-bumps or contour-bumps. Wedge-bumps are typically identifiable by a ramp/tail combination [21], sometimes with flanks, and are dominant in 3D studies. Contour-bumps, alternatively, are usually described by a smooth, continuous surface deformation (relative to wedge-bumps). The efficacy of these geometries is still contested in terms of their performance in drag reduction in pre-buffet conditions; however, both approaches seem to offer either alleviation in 2D/3D [22,23] or complete suppression in 2D [24] in on-design conditions. Given that these devices inherently must be designed for a particular flight condition for optimal performance, the impact of a fixed SCB on a wing can often lead to diminished off-design performance relative to the clean wing configuration. As such, the prospects of a deployable geometry in conditions where they could delay buffet onset or attenuate transonic shock oscillations within the envelope are attractive. Recently, studies by Rhodes and Santer [25,26] showed numerically that morphing SCBs, which can be deployed through the actuation of a flexible membrane, are possible where designs have been generated that offer an optimum trade-off between structural, material, and aerodynamic constraints. Further, research from Jinks et al. [27–30] demonstrated the efficacy of the morphing SCB experimentally and numerically in pre-buffet flows. A limitation of this technique is that under current materials technologies, the sharp geometries required for wedge-bumps (especially in 3D) are not feasible due to sharp corners, requiring large variations in Gaussian curvature over short distances, as well as the additional actuation required.

In the present study, an active shock control system was developed where a contour-based SCB was deployed on the ONERA OAT15A aerofoil within an experimentally observed buffet flow condition. Numerical simulations were performed using URANS to provide an evaluation of an SCB based on a single Hicks–Henne shape function, considering geometry and deployment time. From this, it was observed that the deployment of this type of SCB resulted in either a stable shock solution or a modulation of the self-sustained shock motion, independent of deployment time and short-period transient fluctuations. This measurement was derived from the amplitude of the unsteady lift coefficient obtained, such that sufficiently small peak-to-peak differences were indicative of shock suppression. Based on these observations, a parametric study of the geometric variables defining the SCB was carried out to determine the efficacy of a variety of designs and provide a model for

the dominant features that lead to stable shock configurations. Based on aerodynamic performance metrics, two SCB designs were then evaluated across the numerically observed buffet envelope at $M = 0.73$, where the applicability of this SCB was further explored as a potential shock control device.

## 2. Shock Control Bump Model

### 2.1. Overview of Shock Control Bumps in Transonic Flows

Shock control bumps were originally developed with the intent of controlling static transonic shocks by directing flow outside the shear-layer away from the aerofoil surface. This effect leads to a smearing of the normal shock that sits at the rear of the sonic region, forming the $\lambda$-shock structure, shown in Figure 1, where the total sum of oblique shock-waves within the structure affords better total pressure recovery and hence reduced wave drag [13]. Contingent on the shape and position of the bump, the ideal $\lambda$-shock structure shown in Figure 1 may not be possible, indicating an intrinsic link between the shock-foot location, bump position, and geometry of the bump curvature. Ogawa et al. [21] presented a set of three typical flow structures that exist on a flat plate when the SCB is within the proximity of the shock location, illustrated in Figure 2.

Of these idealised shock systems, the optimum configuration, Type-B, is shown in Figure 2b, whereas Type-A and Type-B (shown respectively in Figure 2a,c) represent off-design cases. Where the stationary shock sits forward of the SCB position, a dual $\lambda$-shock system develops such that the increased curvature due to the presence of the SCB leads to a re-expansion aft of the leading shock leg, thus forming a secondary, smaller $\lambda$-shock system at the SCB crest position. Depending on the distance between the shock and SCB crest, the re-expansion region can remain connected to the supersonic flow region or develop behind a normal shock, resulting in two supersonic regions on the aerofoil surface. For cases where the SCB sits within the sonic region, a re-expansion region will develop at the SCB crest, where the accelerated flow results in a much stronger $\lambda$-shock structure over the tail section of the SCB. The consequence of both of these "off-design" cases is that they typically increase pressure fluctuations in the trailing edge region, with an increase in boundary layer thickness, or total separation.

For transonic shock buffet, SCBs were first explored by Birkemeyer et al. [11], where computations with 2D Reynolds-averaged Navier–Stokes (RANS) and an experiment showed that 2D SCBs could raise the buffet onset boundary to higher angles of attack by introducing a region of re-attached flow in the shear-layer behind the shock rear-leg and the trailing edge. SCBs placed in the shock region did not yield similar results. They argued however that the positioning of such a device for buffet control loses the drag benefits of having the shock placed within the shock region. As such, most of the following work in SCBs for buffet control tended towards more wedge-based geometric configurations. More recently, a study by Tian et al. [24] investigated the performance of a simple hill-type SCB on buffet suppression on the RAE2822 aerofoil at two positions along the rear of the aerofoil. These findings showed that whilst both bumps were capable of mitigating buffet at the design point, the rearward positioned bump provided only a marginal shift in buffet onset. Geoghegan et al. [31] showed that buffet could be controlled for a much wider range of SCB positions for the OAT15A aerofoil in a fixed flight condition.

The consensus on SCB performance, particularly for buffet control, is that the performance of any given bump is largely due to four primary geometric properties; SCB height, SCB length, ramp angle, and SCB position relative to shock position. Under pre-buffet conditions, SCBs have been heavily researched in a wide parametric space involving the geometric variables. There are limited findings on the sensitivity of these ranges within a buffet envelope. Further, the choice of SCB position is made more complex due to the oscillatory nature of the shock within this region. The SCBs investigated by Tian et al. [24] strictly compared two positions aft of the mean shock location with fixed lengths and marginal difference in SCB heights. The work from Geoghegan et al. [31] showed that the range of applicable geometries for buffet suppression is potentially much larger than the limited scope for drag

reduction, which is the case for stationary shock systems. The difficulty in selecting an appropriate geometry now falls on the comparison between resultant steady-state aerodynamic performance relative to a previously unsteady flow-field.

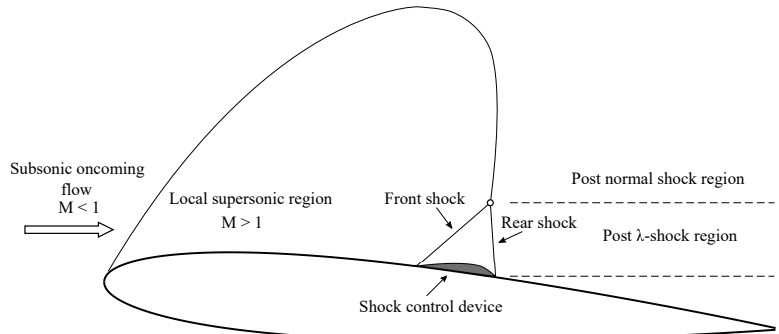

**Figure 1.** Transonic wing section with shock control and $\lambda$-shock structure.

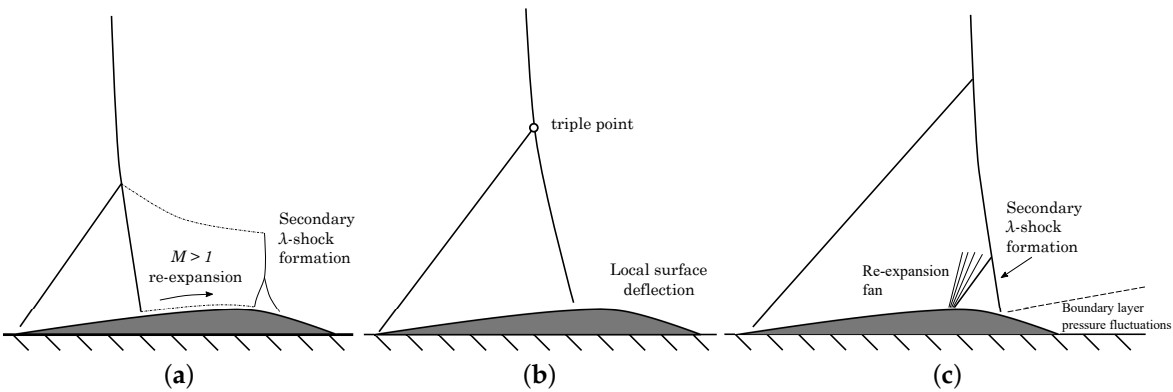

**Figure 2.** Local flow behaviour under shock control bump position relative to shock (adapted from Ogawa et al. [21]. (**a**) Type-A: shock-fore position. (**b**) Type-B: optimum shock position. (**c**) Type-C: shock-aft position.

## *2.2. Shock Control Bump Definition*

The SCB model and deployment technique presented by Geoghegan et al. [31] was used for the study presented in this paper. This model is developed with the concepts of adaptive SCB technology, using a single Hicks–Henne shape-function to define the bump curvature. The shape-function can define a differentiably smooth surface contour based purely on the SCB amplitude, skew, and local coordinate system. The bump width is given as a percentage of chord length, $l_b/c_b$, the position of the bump crest relative to the bump length, $c_b/l_b = 0.5$, yielding a symmetric profile, and the bump local position, $x_s$, which is the distance between the bump crest and the mean shock location, $x_{sh}$. The local position is defined by Equation (1).

$$x_s = \frac{x_0 + c_b - x_{sh}}{l_b} \tag{1}$$

where $x_0$ is the left-hand side starting point of the SCB. The local position coordinate in the function-space, $x_b$, is expressed in Equation (2).

$$0 \leq x_b = \frac{x - x_0}{l_b} \leq 1 \tag{2}$$

The SCB profile is computed using the Hicks–Henne function shown in Equation (3).

$$H(x_b) = \sin^4\left(\pi x_b^m\right), \quad m = \frac{\ln 0.5}{\ln(c_b/l_b)} \tag{3}$$

Since $H(x_b)$ is normalised by definition, the bump height, $h_b$, can be applied directly by scaling the function. An illustration of the SCB profile function is shown in Figure 3 with its size and relative position on the OAT15A. The deployment of the bump function to the aerofoil surface is modelled through a sinusoidal ramp function with ramp frequency, $f_r$, and a unit step function, $u(t)$, to control activation/deactivation, as shown in Equation (4).

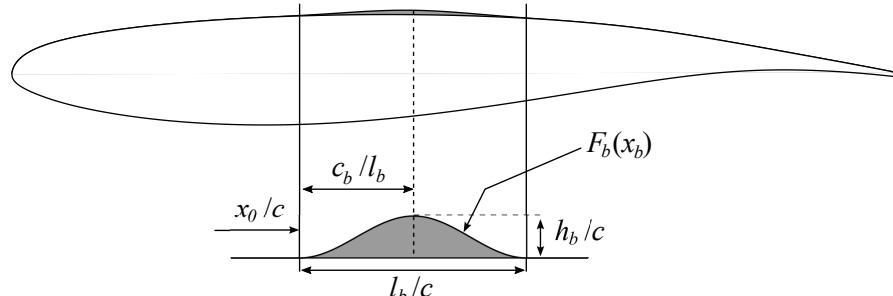

**Figure 3.** Shock Control Bump (SCB) model definition relative to the OAT15A aerofoil.

$$T(t) = \left[ u(t - t_{deploy}) - u(t - t_{stop}) \right] \sin\left(2\pi f_r \bar{t}\right) \tag{4}$$

The dynamic control function implemented was developed with the capability of augmenting the deployment speed, $f_r$, as well as the onset time, $t_{deploy}$, such that the bump amplitude could be adjusted actively. For this study, only deployments up to a maximum amplitude were explored. The deactivation time, $t_{stop} = t_{deploy} + t_{peak}$, was defined such that surface deformations were halted once the bump was at peak amplitude. Given that the ramp function was sinusoidal, the peak time was defined as $t_{peak} = 1/(4f_r)$. The adjusted time, $\bar{t}$, was determined such that the ramp function advanced relative to the position of the bump and was incremented at the same rate as the flow time. The combination of Equations (3) and (4) with bump height modulation is given in Equation (5).

$$F_b(x_b, t) = h_b T(t) H(x_b) \tag{5}$$

Numerically, the SCB deployment was modelled using surface augmentation and was performed using a diffusion-based mesh motion, where the diffusion coefficient was based on the boundary distance. This method ensured that the mesh integrity was maintained in the near-field deformation region and that the far-field cells were largely unaffected. Figure 4 shows a comparison of the grid topology local to the SCB before and after deployment.

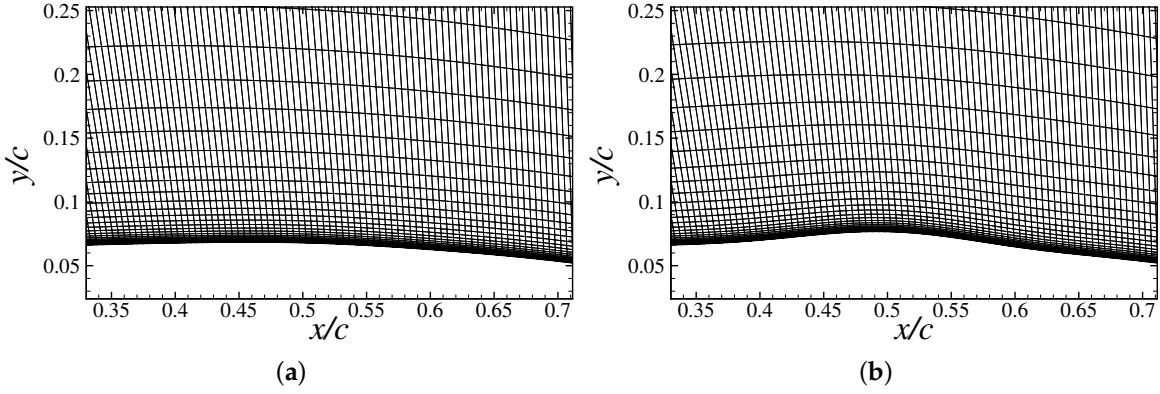

**Figure 4.** Near-field mesh deformation during SCB deployment. (**a**) Pre-deployment. (**b**) Post-deployment.

## 3. Validation of Unsteady Reynolds-Averaged Navier–Stokes Simulations

### 3.1. OAT15A Experimental Test Case

The SCB performance was evaluated on the ONERA designed OAT15A supercritical aerofoil, which has been experimentally observed to undergo buffet under certain flow conditions. Experiments on this section were performed in the S3ChContinuous Research Wind Tunnel at the ONERA Chalais-Meudon Center, detailed by Jacquin et. al. [3]. A wind tunnel model of 12.3% relative thickness, 230 mm chord, 780 mm span, and a 1.15 mm thick trailing edge was constructed for the experiment. The model ensured a fixed boundary layer transition at 7% chord through the installation of a carborundum strip on the upper and lower surfaces. The experiments were carried out using a model outfitted with 68 static pressure sensors and 36 unsteady Kulite pressure transducers. The investigation served as an excellent baseline for numerical evaluation as the experiment provided a variety of visualisation techniques to capture the bulk flow phenomenon and nuanced flow features within the shock excursion, as well as the mean and root-mean-squared (RMS) pressure statistics. The test consisted of an angle of attack sweep at $M = 0.73$ to obtain data for buffet onset, as well as Mach number sweeps at $\alpha = 3°$ and $\alpha = 3.5°$. In this study, the data at $M = 0.73$ and $\alpha = 3.5°$ were used to validate the numerical approach.

### 3.2. CFD Model

Independent computational studies successfully have captured the inherent flow characteristics of transonic shock buffet using URANS modelling; however, prominent sensitivity to a variety of simulation variables has been observed [32–39]. The most pertinent of these was the choice of turbulence closure [35,40–42], grid resolution local to the shock travel region [36,43,44], and the use of dual time-stepping with acoustic temporal resolution [43,45]. The value of URANS simulations in predicting this flow feature lied in the fact that the low frequency characteristics of oscillation were present at comparatively longer time-scales than shear-layer eddies [46].

Simulations performed in this study used the commercial, cell-centred finite volume CFD code, ANSYS Fluent [47]. The 2D implicit density-based solver was used to formulate coupled continuity/momentum/energy equations. The governing equation for a single-component fluid, which describes the mean flow properties, in integral Cartesian form for a control volume $V$ with differential surface area $d\boldsymbol{A}$ is given in Equation (6). This model used the dual-time formulation, which introduced a preconditioned pseudo-time-derived term to the baseline density-based vector form of the Navier–Stokes equation.

$$\frac{\partial}{\partial t} \int_V \boldsymbol{W} dV + \Gamma \frac{\partial}{\partial \tilde{t}} \int_V \boldsymbol{Q} dV + \oint [\boldsymbol{F} - \boldsymbol{G}] \cdot d\boldsymbol{A} = \int_V \boldsymbol{H} dV \tag{6}$$

where,

$$\boldsymbol{W} = \begin{bmatrix} \rho \\ \rho u \\ \rho v \\ \rho E \end{bmatrix}, \quad \boldsymbol{Q} = \begin{bmatrix} p \\ u \\ v \\ T \end{bmatrix}, \quad \boldsymbol{F} = \begin{bmatrix} \rho v \\ \rho v u + p\hat{\boldsymbol{i}} \\ \rho v v + p\hat{\boldsymbol{j}} \\ \rho v E + p v \end{bmatrix} \quad \boldsymbol{G} = \begin{bmatrix} 0 \\ \tau_{xi} \\ \tau_{yi} \\ \tau_{ij} v_j + q \end{bmatrix}$$

The vector $\boldsymbol{H}$ contains source terms. The variables, $\rho$, $\boldsymbol{v}$, $E$, and $p$ represent the density, velocity, total energy per unit mass, and fluid pressure, respectively. The differential operators were taken with respect to the physical time, $t$, and the pseudo-time, $\tilde{t}$. The viscous stress tensor was given by $\boldsymbol{\tau}$ and heat flux by $\boldsymbol{q}$. The unsteady preconditioning matrix, $\Gamma$ was used in this formulation to improve the scaling of artificial dissipation and to optimise the number of sub-iterations required at each physical time step.

Inviscid fluxes were resolved using an upwind Roe flux difference splitting algorithm with a blended central difference/second order upwind MUSCL scheme to extrapolate convective quantities. Diffusive fluxes were treated using second-order central differencing method. The convective and diffusive gradients were constructed via a cell-based least-squares approach and completed by the Gram–Schmidt decomposition of the cell coefficient matrix. Menter's $k - \omega$ SST [48] turbulence model was used for closure of the Navier–Stokes equations, with all turbulent convective quantities solved through second-order upwind differencing and isolated from the coupled continuity/momentum/energy equations.

The solution domain consisted of a 2D CH-type structured mesh, with far-field boundaries at 80 chord length spacing from the profile, shown in Figure 5, where the far-field mesh is shown in Figure 5a and near-field in Figure 5b. The domain was subdivided into two zones: laminar region upstream and up to 7% of the aerofoil chord section; and a turbulent zone for the remainder. This treatment was performed in order to replicate the experimentally imposed boundary layer transition. The grid size and time-step employed in this paper were determined through the spatial and temporal refinement and simulation sensitivity studies provided in Giannelis et. al. [39,49]. The spatial domain was comprised of approximately 48,000 grid points, whereas a non-dimensional time-step of $\Delta \bar{t} = 0.01$ sufficiently yielded temporally converged solutions.

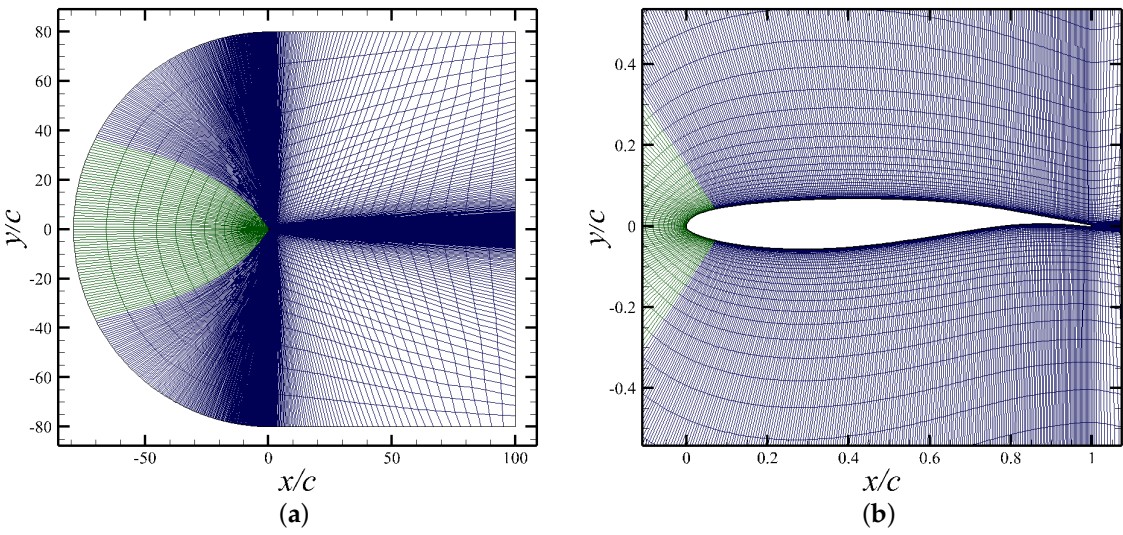

(a)    (b)

**Figure 5.** Computational grid; green: laminar zone, blue: turbulent zone. (**a**) Far-field grid topology. (**b**) Near-wall grid topology, Grid C.

From the unsteady simulations, a summary of the pressure statistics of the transient numerical analysis is shown in Figure 6 with comparison to the experimental data for the present test case. Figure 6a shows that there was exceptional agreement with the experimental mean pressure coefficient. Figure 6b shows the RMS pressure coefficient, which demonstrated that whilst the RMS pressure amplitude and trailing edge pressure fluctuations were marginally under-predicted, the shock travel and mean location of the shock were reasonably captured. The mean shock location for the test case, at $M = 0.73$ and $\alpha = 3.5°$, was computed at 45% of the chord.

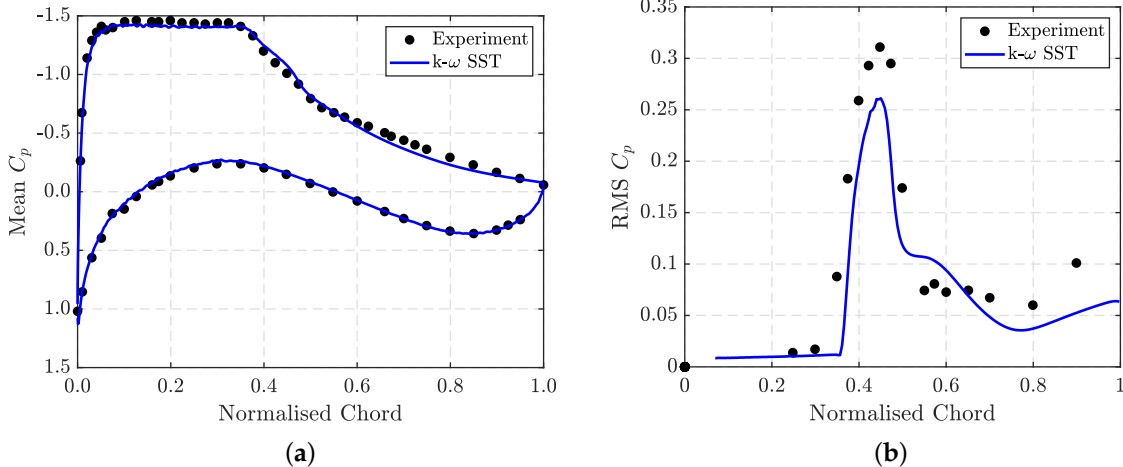

**Figure 6.** CFD computed pressure statistics comparison with Jacquin et al. [3] (**a**) Mean pressure coefficient. (**b**) RMS pressure coefficient.

## 4. Influence of SCB Parameters on Shock Control and Flow Response

A sensitivity study of the parameters that formed the basis of the present SCB model was performed in order to determine the critical geometric and temporal parameters that led to buffet suppression. The geometries evaluated in this study were strictly symmetric profiles, such that an SCB crest location of $c_b/l_b = 0.5$ was preserved across all test cases. Table 1 contains a summary of the four test cases where individual parameters were varied. These values were based on the ranges cited in a review paper by Bruce [50] on general SCBs, the geometries evaluated by Tian et al. [24,51], and those from Geoghegan et al. [31].

**Table 1.** Sensitivity study SCB cases and associated parameter values.

| Case | Fixed Parameters | Active Parameter(s) |
|------|------------------|---------------------|
| Case 1 | $x_{sh}/c = 0.45$, $l_b/c = 0.4$, $h_b/c = 0.005$ | $f_r = 1 : 50$ Hz, $x_s/c = 0.05, 0.1$ |
| Case 2 | $x_{sh}/c = 0.45$, $l_b/c = 0.4$, $h_b/c = 0.005$, $f_r = 50$ Hz | $x_s/c = -0.1 : 0.15$ |
| Case 3 | $x_{sh}/c = 0.45$, $l_b/c = 0.4$, $x_s/c = 0.0$, $f_r = 50$ Hz | $h_b/c = 0 : 0.015$ |
| Case 4 | $x_{sh}/c = 0.45$, $h_b/c = 0.005$, $x_s/c = 0.05$, $f_r = 50$ Hz | $x_s/c = -0.1, 0.15$ |

It was observed that several of the geometries in the cases presented in Table 1 resulted in a termination of the inherent shock oscillation, and this section will compare the lift coefficient time histories, mean and peak lift values, mean pressure coefficients, and lift-to-drag ratios as further assessments of the SCB performance. This provided the basis for further parametric studies, as well as helping to identify the optimum configuration for design point buffet suppression and restoration of favourable aerodynamic performance.

### 4.1. Case 1: Impact of Deployment Frequency Variation

Given that the purpose of the SCB design presented in this paper was to activate within or near the onset of transonic shock buffet, it was necessary to first evaluate the impact that deployment speed had on the resultant flow-field, as well as any transients that may be present during the perturbation. The time-varying component of the SCB deployment function in Equation (5) was based on a quarter period sine-wave, which therefore meant that the rate of change in surface curvature was non-linear and was largest at the start and end of the deployment phase. By maintaining a constant amplitude, the effect of deployment rate could be demonstrated by varying the ramp frequency. Figure 7 shows a comparison of the transients generated by SCB activation on the test case, where Figure 7a,b shows the

solution for an SCB at an "ideal" position and a position at the cusp of buffet onset, respectively. The lift history is presented with respect to $\bar{t}$, the simulation time in seconds relative to bump deployment; hence, $\bar{t} < 0$ represents the OAT15A at the test flight condition without control.

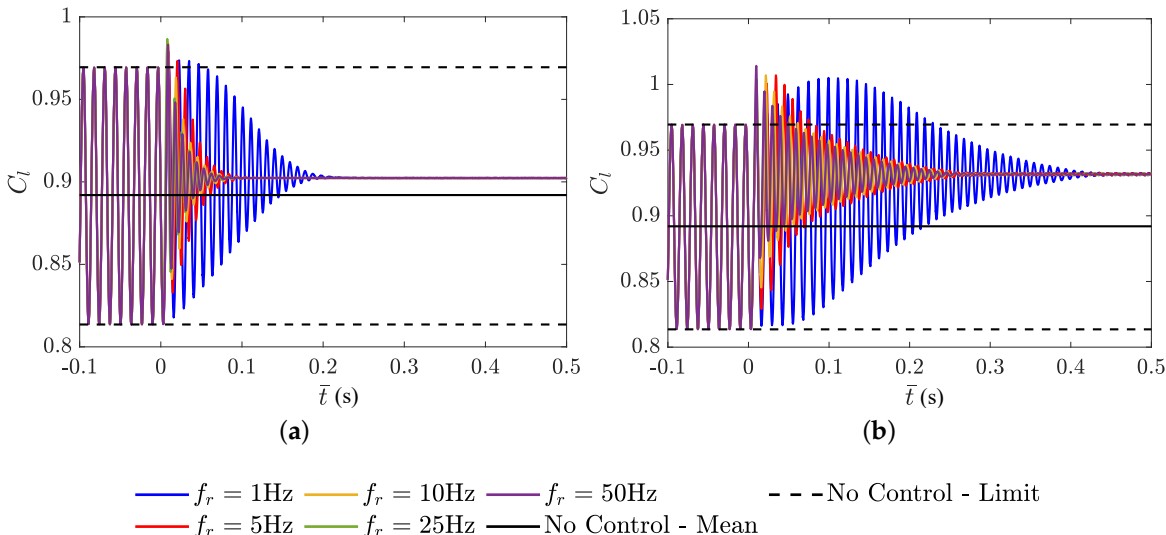

**Figure 7.** Lift coefficient time-history for Case 1 compared to the mean and peak values without control. (**a**) $x_s/c = 0.05$. (**b**) $x_s/c = 0.10$.

It is immediately apparent that for frequencies of $f_r = 10$ Hz and above, the flow-field reached a stable steady-state where all shock oscillation was arrested within 10 cycles. In both cases, there was considerable overlap between $f_r = 25$ Hz and $f_r = 50$ Hz. Further, the ramp frequency had no effect on the lift coefficient of the new flow state, but rather the time taken to reach it, as the settling time for $f_r = 1$ Hz was $t_{s.t.} \approx 0.2$ s and $t_{s.t.} \approx 0.42$ s for Cases 1(a) and 1(b), respectively, compared to the higher ramp frequencies where the time-scale was approximately half. Observing the transient oscillations between the two positions also showed that as the SCB was moved further away from the mean shock location, the time to reach steady-state was increased, and damping decreased. This trend was also observed by Geoghegan et al. [31] where this increase in settling time was indicative of the SCB position approaching a critical point where shock oscillations would re-establish on the aerofoil surface. Further, the overshoot present at activation was more pronounced at higher frequencies, though it was primarily dependent on the instantaneous position of the rear shock leg at the point of activation relative to the steady-state position. The relationship between SCB relative position and resultant lift was explored further in Case 2. The key observation from varying ramp frequency was that it ultimately did not affect the suppression of shock oscillations or alter the steady-state flow-field. The implication of this on a practical level was that a device that could deploy the SCB almost instantaneously would yield fewer shock cycles before termination, but was dependent on mechanical and material constraints. For the remainder of this study, a ramp frequency of $f_r = 50$ Hz was used as it led to a converged solution faster than at lower frequencies.

*4.2. Case 2: Impact of the SCB Position Relative to the Mean Shock Location*

It is well understood that for hill-type SCBs (as well as wedge-type) such as the present model, there is a strong relationship between crest position and aerodynamic performance. Generally speaking, the research has tended to suggest that rearward positioned bumps are far more preferable for flow control in static shock systems [50,52]. This dependence was explored under a static shock context by Tian et al. [53], whereby lift-to-drag ratios were improved markedly at SCBs positioned up to 30% aft of the shock location. However, given that onset can often be brought on earlier than in no-control cases, much of the existing literature only considered flight conditions up to the point at

which on-set was reached, but was not applicable within the natural buffet envelope. In more recent literature, Tian et al. [24] showed that contour-based SCBs positioned between 10 and 18% aft of the mean shock location were able to damp out shock oscillations in a buffeting flow-field. Additionally, Geoghegan et al. [31] showed that for the OAT15A aerofoil, buffet suppression existed for a much wider range of positions including SCBs positioned at and in front of the mean shock location. Case 2 explored the sensitivity of SCB position relative to the mean shock location for the test case. Figure 8a shows the mean lift coefficient measured, with boundaries indicating the maximum and minimum lift coefficient variation over a period of 2 s after SCB deployment. The results revealed that there existed, at least for a bump with 0.5% chord crest height, a wide range of applicable placements that resulted in the termination of transonic shock oscillation. Further, the generally agreed on notion that aft-positioned SCBs were ideal did in fact have a limit, whereby there existed a limit on the aft positioning after which the regeneration of shock oscillation developed. This resurgence in shock oscillation presented at $x_s/c = 0.11$, after which the buffet was amplified with increasing distance.

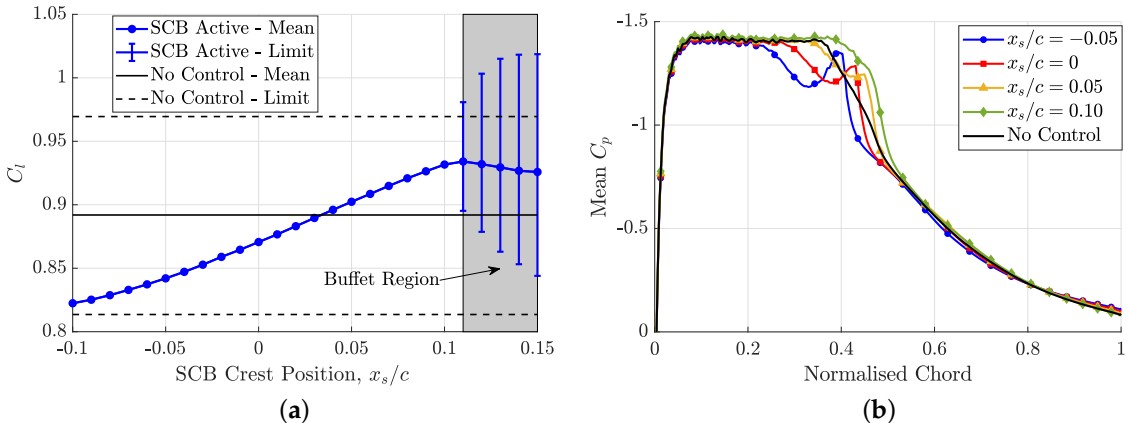

**Figure 8.** Lift and pressure statistics for Case 2. (**a**) Mean and peak lift. (**b**) Mean pressure coefficient.

Given that the primary objective of the deployed SCB was achieved for most geometries in Case 1, i.e., a stable shock was formed within a buffeting flow-field, it was necessary to consider the secondary objectives to establish an optimum geometric configuration. It was evident from Figure 8a that the lift coefficient was increasing monotonically for positions between $x_s/c = -0.1 : 0.1$; however, for values less than $x_s/c = 0.04$, the resultant lift sat below that of the mean value from the uncontrolled state. The impact of these SCB geometries is highlighted further in Figure 8b, which shows the mean pressure coefficients at $x_s/c = -0.05$, 0, 0.05 & 0.1. The two extremes presented here at positions $x_s/c = -0.05$ and at $x_s/c = 0.1$ demonstrated the least favourable designs. For the SCB at $x_s/c = -0.05$, there was a strong deceleration of the flow in the supersonic region starting at the foot of the ramp, followed by an expansion fan that re-accelerated the flow, resulting in a local pressure drop at the bump crest near the pressure rooftop, which terminated in a strong shock immediately aft. The apparent loss of lift could be directly attributed to the variation in local surface Mach number in this region. Alternatively, at the furthest aft pre-buffet position, $x_s/c = 0.1$, the resulting shock sat slightly behind the mean position before control; however, the smearing of the terminating normal shock caused by the typical $\lambda$-shock structure was not prominent. It is important to note here, that whilst there was some small variation in maximum/minimum lift at this position, it was primarily due to fluctuation in the shear-layer aft of the SCB, not shock motion, and hence was regarded as being on the cusp of shock oscillation. The SCBs at $x_s/c = 0$ and 0.05 represented the ideal pressure distribution and followed similar profiles to those cited by Jinks et al. [30]. In these cases, the presence of the bump only partially re-accelerated the sonic region or, in the case of $x_s/c = 0.05$, showed a smooth near monotonic decrease in pressure drop, before the weaker rear shock leg was present. The advantage of the more aft of these two geometries

was that $\lambda$-shock led to a lift coefficient that was marginally increased compared to the mean lift of the baseline aerofoil.

The nature of the shock stabilisation is further illustrated in Figure 9, where Mach contours are shown with streamlines in the boundary layer region aft of the SCB for the four positions highlighted in Figure 8b. The Mach contours served to re-enforce the observation of local Mach number deceleration and re-acceleration in the vicinity of the SCB. It was clear that for Case 2(a), the supersonic region underwent a brief deceleration at the ramp of the SCB, a consequence of it being further forward with respect to the stabilised shock location, and bore a strong resemblance to the typical adverse shock Type-C structure (shown in Figure 2c). As the SCB was moved rearward, this expansion fan was weakened, until at Case 2(c), where there existed a coherent $\lambda$-shock without any large perturbation in Mach number before the terminating shock. The $\lambda$-shock structure compressed considerably in Case 2(d), where there was a very large gradient in local Mach number. The velocity streamlines demonstrated a phenomenon where local surface curvature, a result of the combination of the SCB and the intrinsic curvature of the aerofoil surface, led to a dual re-circulation system in the boundary layer aft of the bump. The primary cell sat at the trailing edge of the aerofoil and was persistent across all cases, including under buffeting flows (however, the size varied in these cases), and the secondary cell developed in the tail region of the SCB. The development of these two re-circulation cells and the interactions between them appeared to be a primary mechanism in the suppression of buffet. In Case 2(a), the secondary re-circulation cell developed underneath the shock foot, within a relative concavity produced between the SCB crest and the aerofoil surface. It interacted weakly with the primary cell, leading to two clearly defined zones. As the SCB was moved aft, the concave surface curvature was slowly reduced, moving the secondary cell rearward and increasing its span-wise aspect ratio. This reduction in curvature also promoted the interaction between the primary and secondary cells seen most prominently in Case 2(d), just before buffet on-set. For cases exceedingly aft of that for Case 2(d), the two re-circulation cells merged to produce a fully reversed flow between the shock and trailing edge, re-introducing the shock oscillation mechanism.

The size and length of the re-circulation cell variation with SCB position could be further explored through the presentation of normalised mass flow rates with respect to wall normal distances. Figure 10a illustrates the overall variation of the velocity for the boundary layer in the aft 50% of the aerofoil chord. The effect of re-circulation cell merging, a consequence of positioning the SCB further aft, rendered itself as a slight reduction in the boundary layer height. Figure 10b shows the same profile, focused on the variations between a wall normal distance of $0 - 1\%$ chord, such that the reversed flow present due to the re-circulation cell system can be more clearly seen. For the SCB with $x_s/c = -0.05$, the post bump crest re-circulation cell appeared distinctly, as the reversed flow region stopped between $x/c = 0.65 - 0.70$, and the trailing edge cell developed at $x/c = 0.80$. As observed with the streamlines in Figure 9, moving the SCB further rearward promoted the interaction between the primary and secondary re-circulation cells, hence the development of full flow reversal across the entire rear portion of the aerofoil. Nonetheless, the amplitude of this persistent reversed flow region was considerably small between the two cells. The break-up of the large separated flow region (that would otherwise be present on the clean aerofoil) was suggested as a method of offsetting the buffet boundary by inhibiting communication between the aerofoil trailing edge and shock motion [11,54] and was observed here to eliminate buffet under active SCB deployment.

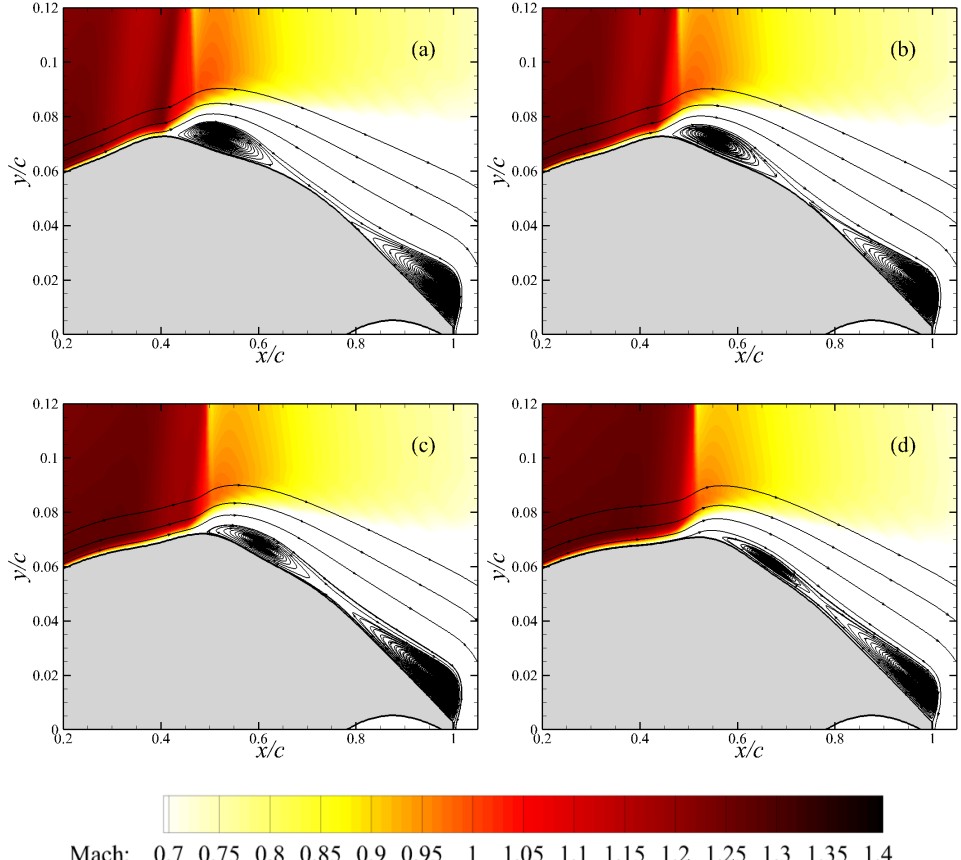

**Figure 9.** Mach contours and streamlines for (**a**) $x_s/c = -0.05$, (**b**) $x_s/c = 0$, (**c**) $x_s/c = 0.05$, (**d**) $x_s/c = 0.10$.

### 4.3. Case 3: Impact of SCB Crest Height

In consideration of the design and efficacy of 2D SCBs for stationary shock control, the height of the bump is as critical a factor as placement on the aerofoil surface. There is a consensus in the literature that SCBs with heights approximately equal to the incoming boundary layer, particularly for static shock weakening, yield the ideal flow control compared to taller bumps [50]. This sizing constraint is more difficult to achieve when considering SCBs already within a buffeting flow as the height and extent of the boundary layer varies across a shock oscillation cycle. More recently, Mayer et al. [22] suggested that in terms of SCBs (wedge-type) for buffet control, the buffet behaviour was relatively insensitive to bump crest height. Further, Tian et al. [24] demonstrated buffet suppression in 2D with contour like SCBs using heights of $h_b/c = 0.008$ and $0.01$; however, there are no definitive relationships between bump crest height and buffet suppression performance. In this section, the influence of crest height is explored against an SCB position where TSB was shown to be eliminated. Figure 11a shows the mean and peak lift characteristics for Case 3, where it is clear that beyond a crest height of $h_b/c = 0.002$, all shock oscillations (and large shear-layer fluctuations) were damped out completely. The lift coefficient variation with crest height exhibited two distinct regions, a relatively linear range between $h_b/c = 0.002 : 0.012$ and a discontinuous drop region between $h_b/c = 0.013 : 0.015$.

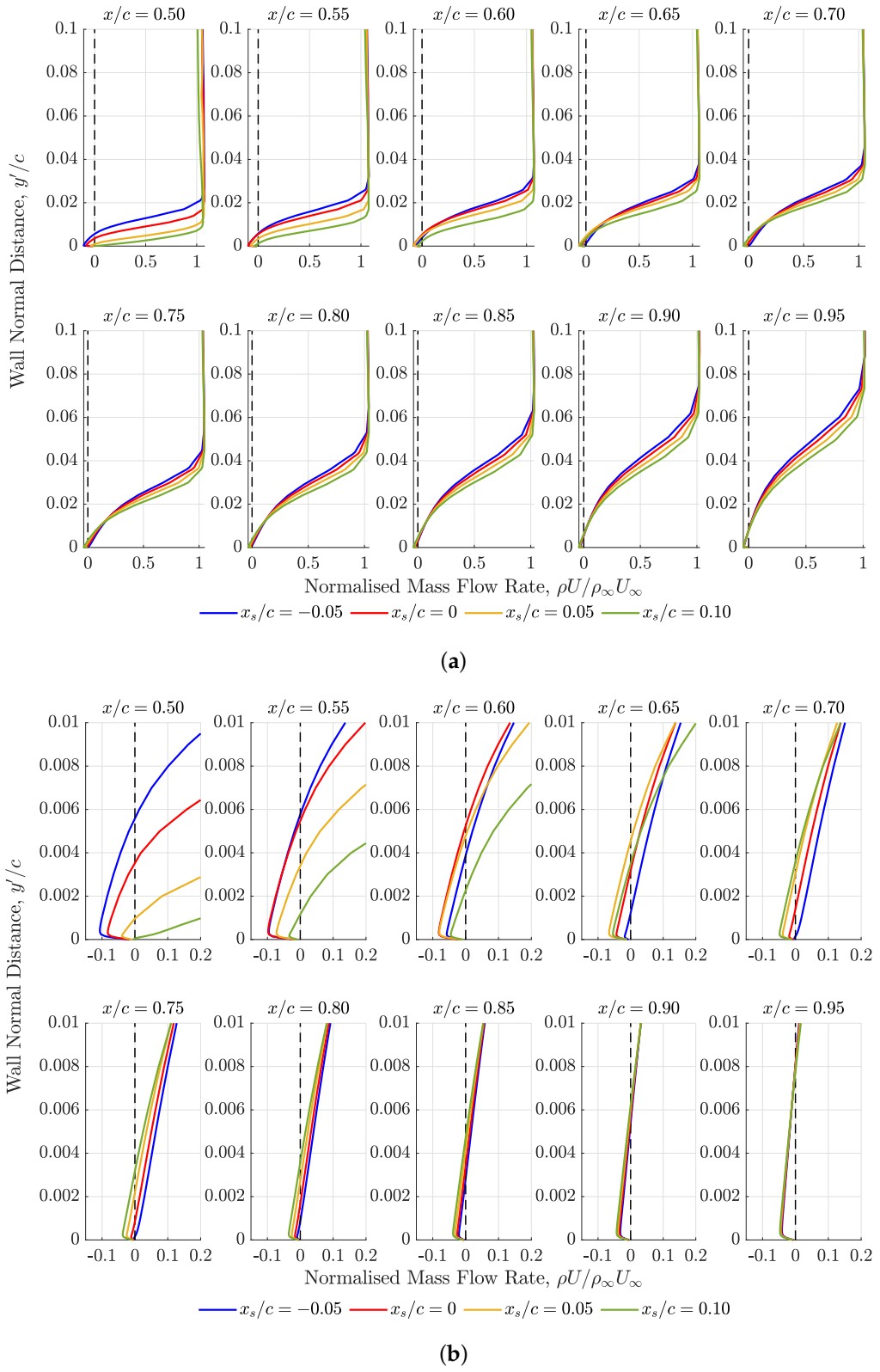

**Figure 10.** Comparison of normalised mass flow rate for Case 2 SCB designs. (**a**) Profile at 10% chord wall normal distance. (**b**) Profile at 1% chord wall normal distance.

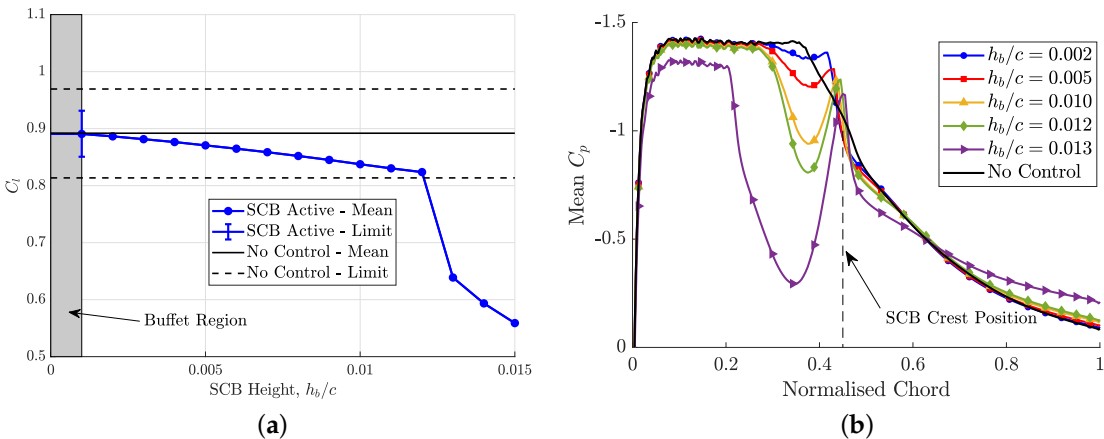

**Figure 11.** Lift and pressure statistics for Case 3. (**a**) Mean and peak lift. (**b**) Mean pressure coefficient.

Figure 11b shows the evolution of the mean pressure coefficient with respect to SCB crest height, indicating that between $h_b/c = 0.002 : 0.005$, the SCB maintained a similar profile to the ideal pressure distributions seen with variation in positions in Case 2. Two interesting phenomena developed as crest height was increased, which could be identified through the pressure coefficients: (1) as crest height increased, the rear terminating shock moved aft towards the bump crest position, and (2) the supersonic deceleration and re-expansion region due to the SCB ramp angle was increased up to a point where, at $h_b/c = 0.013$, there were two normal shock systems present on the aerofoil surface.

Figure 12 illustrates the typical variation in Mach contours across the linear region. The structure of the shock at the SCB crest height, $h_b/c = 0.002$, resembled the ideal Type-B $\lambda$-shock structure (shown in Figure 2b) with a consistent smearing of the normal shock at the foot. However, it maintained a reasonably strong deceleration gradient at the front leg. Figure 12b,c shows that the shock structure local to the SCB transitioned from Type-C to Type-A, where the rear-leg of the primary $\lambda$-shock was driven forward and partially disconnected from the normal shock at the rear-leg of the secondary $\lambda$-shock. These two regions were connected through a "supersonic tongue", preventing the flow from fully decelerating. This result was likely a consequence of the geometric definition of the present SCB. The leading ramp angle, whilst non-constant due to the Hicks–Henne function, increased in rate sooner for taller SCBs given that the length parameter was fixed. This presented a trade-off in the geometry as only two of the three variables could be controlled for a symmetric SCB.

Figure 13 illustrates the Mach contours in the discontinuous drop-off region of the lift performance. The Mach variation across the change in crest height from $h_b/c = 0.012$ to $h_b/c = 0.013$ showed a total splitting of the $\lambda$-shock structure, resulting in a separate dual shock system of which neither exhibited the shock smearing. Whilst these amplitudes still resulted in a stable shock system, the flow and aerodynamic characteristics were detrimentally affected. The high flow turning angle aft of the bump crest at the critical crest height led to the dual re-circulation cells merging and forming fully reversed flow within the shear-layer, something that is characteristic of buffet onset; however, the flow remained in a steady-state. Due to the nature of the shock structure here, it was unlikely that the typical TSB mechanism would develop, as the dual shock system was significantly weaker than the system that was present without control.

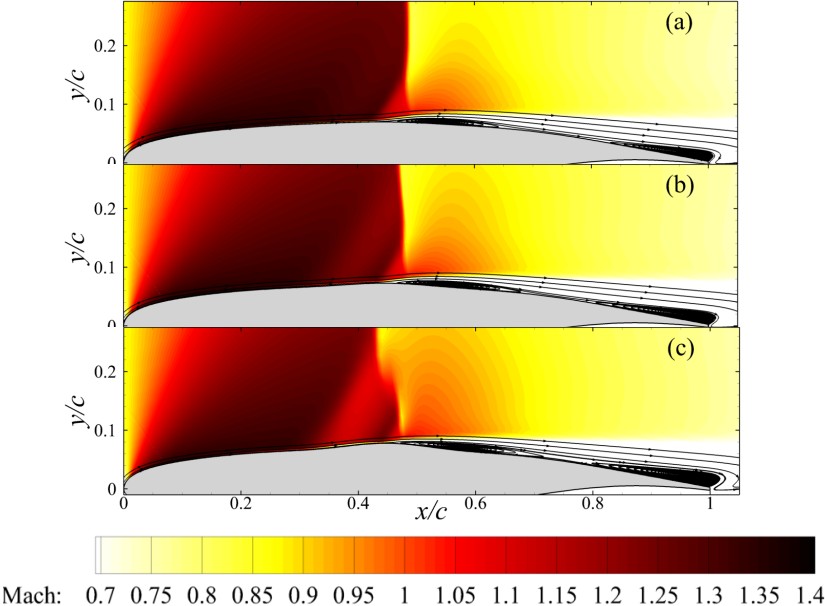

**Figure 12.** Mach contours and streamlines of the local flow-field for height varying SCB. (a) $h_b/c = 0.002$ (b) $h_b/c = 0.005$ and (c) $h_b/c = 0.010$.

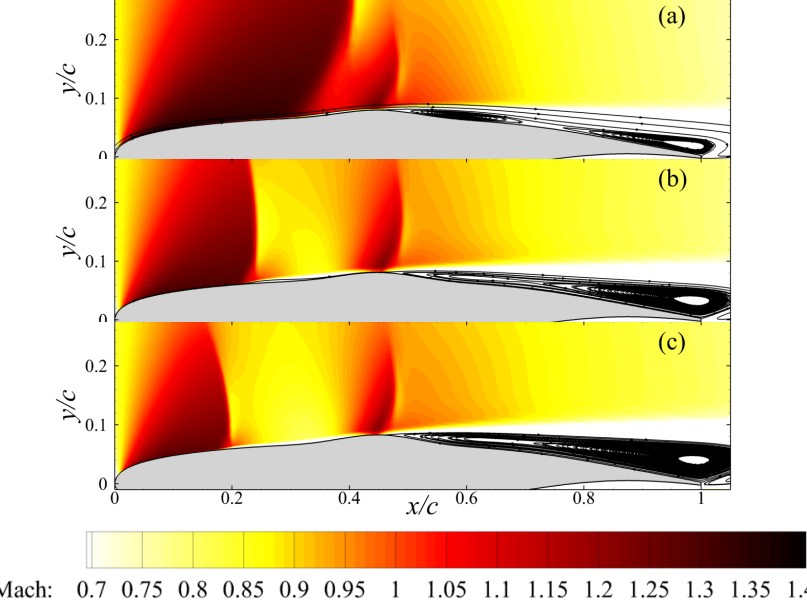

**Figure 13.** Mach contours and streamlines of local flow-field for large SCB heights. (a) $h_b/c = 0.012$, (b) $h_b/c = 0.013$ and (c) $h_b/c = 0.015$.

### 4.4. Case 4: Impact of SCB Length

In SCB design, consideration must be given to the length of the bump in addition to its position and height. Typically, in SCB designs, a ramp/tail angle is defined as a control variable, whereas in the current model, there is no explicit control of this variable. Given that there exists an inherent relationship between these three parameters and the ramp/tail angle of the design, it is necessary to analyse the impact of variation in SCB length relative to fixed crest height and position. Further constraints on the bump length include: limitations on the extent such that the parametric equation leads to a profile that exceeds the aerofoil chord length; and physical limitations for the space needed to implement such a device. With reference to the RAE2822 aerofoil, Jinks et al. [30] presented the design

of a contour-type actuated SCB with $l_b/c = 0.2$ for adaptive static shock control, whereas Tian et al. [24] implemented an SCB with length $l_b/c = 0.4$ in their study of buffet suppression. The former of these two studies used a two-point control system for actuation and hence had much more control over the ramp angle, which is likely an incentive for a narrower profile. The latter implemented a similar geometry to the design in this study and suffered the same limitation of a single peak with symmetric ramp and tail curvature, which lent itself towards a longer SCB extent. The present analysis hence considered the performance of a fixed geometry SCB with lengths varying from $l_b/c = 0.2 : 0.45$ to identify if there existed a strong impact on the ability for this SCB design to suspend shock oscillation and subsequently its independent effect on the aerodynamic performance.

Figure 14a,b shows the resultant lift coefficient of the SCB with varying length and mean pressure coefficient at key positions. In all lengths considered within the parameter range, the buffet was completely suppressed, which given that the other SCB shape variables were chosen based on a known buffet control geometry, was not surprising. There appeared to be a weak interaction between the length and overall lift coefficient such that increasing the length led to a decrease in lift given all other variables were held constant; however, this margin was within $\Delta C_l = 0.01$ from $l_b/c = 0.2 : 0.4$. When compared to the effects of varying position and height, it appeared that both the primary and secondary performance metrics were relatively unaffected. Regarding the pressure coefficients shown in Figure 14b, it was clear that the slight variation in lift performance was a result of the shock smearing within the $\lambda$-shock region with increasing length, as well as the strength of the secondary re-circulation cell that formed at the foot of the normal shock. For the longer SCBs, there was a more favourable pressure gradient introduced by the smaller ramp curvature and the presence of altered surface curvature earlier in the sonic region. This also led to a narrower secondary re-circulation cell due to the smaller flow turning angle behind the SCB crest.

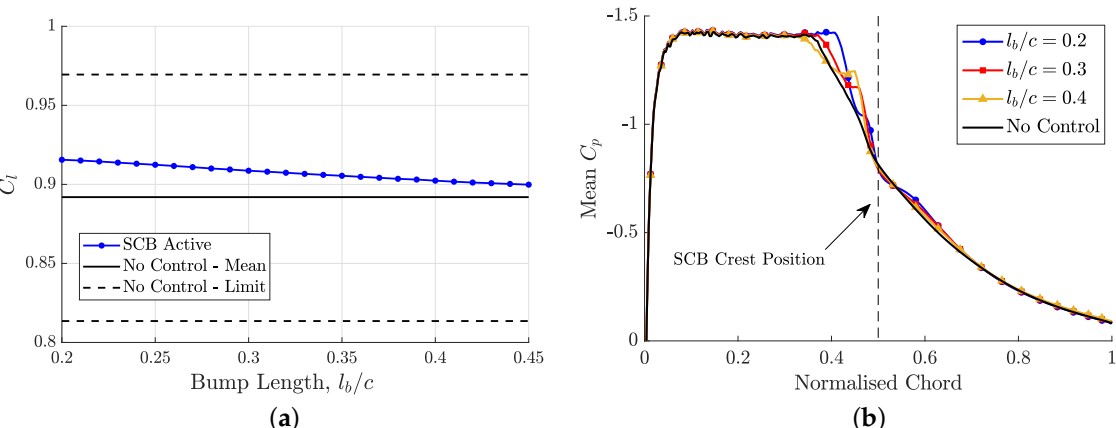

**Figure 14.** Lift and pressure statistics for Case 4. (**a**) Mean and peak lift. (**b**) Mean pressure coefficient.

Figure 15 further illustrates this phenomenon through Mach number contours local to the SCB position. As the SCB with length $l_b/c = 0.2$ had a comparatively large ramp angle, the supersonic region underwent a short Mach deceleration before the terminating shock, which presented as the typical Type-B $\lambda$-shock structure. Increasing the length led to a widening of the deceleration region and reduction in the Mach gradient at the front-leg of the $\lambda$-shock, hence providing better pressure recovery aft of the rear-leg normal shock.

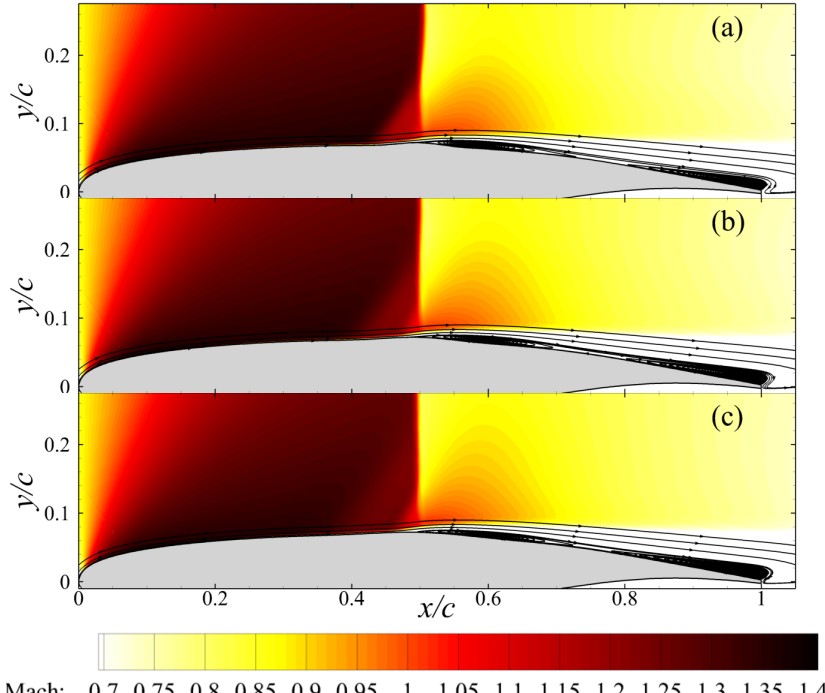

**Figure 15.** Mach contours and streamlines of the local flow-field for length varying SCB. (**a**) $l_b/c = 0.2$, (**b**) $l_b/c = 0.3$ and (**c**) $l_b/c = 0.4$

There appeared to be a balance here between the strength of the terminating normal shock and the interaction between the dual re-circulation cells that maintained the buffet suppression characteristics observed in the sensitivity analysis. Given the relationship between ramp angle with length and height, it was hypothesised that further increasing the SCB length beyond $l_b/c = 0.45$ (or the crest position was moved further forward) would result in a reappearance of buffet as there would be insufficient introduced curvature to promote the development of the dual re-circulation system. Further, for bumps narrower than $l_b/c = 0.2$, the same discontinuous drop in lift observed in Figure 11a would likely present as the SCB would effectively resemble a wall separating the reconnected re-circulation zones with a strong normal shock system that was held in place by the expansion fan encouraged by the sharp curvature. Given that a stronger relationship between lift performance and SCB height was observed, the following sections explore the effect of height coupled with position. A wider bump is preferable despite the relatively minor decrease in lift coefficient when compared to the narrower bumps. Thus, the remainder of the study will consider only SCBs with length $l_b/c = 0.4$.

## 5. Buffet Suppression Envelope

Based on the sensitivity of the SCB geometric variables to buffet suppression and mean aerodynamic performance, this section presents a full parametric analysis of the effects of SCB crest position and height with reference to the baseline test case in order to assess the relationship between the two variables. The parametric space consisted of a range of $x_s/c = -0.1 : 0.15$ with crest heights ranging from $h_b/c = 0.001 : 0.01$, whilst length $l_b/c = 0.4$ and ramp frequency $f_r = 50$ Hz were held constant. In order to evaluate the efficacy of any given SCB design within the design space, the following three criteria must be met:

1.  Shock oscillations at the design point must be suppressed
2.  The resultant steady lift coefficient should not be less than the mean lift coefficient of the uncontrolled aerofoil in the test case.

3.  The induced augmentation to the boundary layer aft of the SCB and total pressure recovery should not result in a diminished lift-to-drag ratio relative to the mean performance of the uncontrolled aerofoil in the test case.

Of these criteria, 2 was established to prevent the pitfall of some control devices, where the buffet onset boundary was raised to a higher incidence angle, but resulted in a smaller lift coefficient such that the design $C_l$ still sat within the buffet envelope. Criterion 3 was defined such that if the SCB was capable of quenching transonic shock oscillations, its performance was evaluated in a similar context to those of SCBs that aimed to improve pressure recovery and reduce wave drag.

An evaluation of Criteria 1 is illustrated in Figure 16 where the quasi-steady peak-to-peak lift coefficient difference (calculated once all initial transient oscillations decayed) is shown and reveals that there was in fact a large parameter space within which transonic shock oscillations were completely damped out. The threshold for buffet suppression was considered as having a peak-to-peak lift difference of $\Delta C_l < 0.02$, as oscillations within this bracket, if present at all, were primarily due to fluctuations in the shear-layer (especially with interacting dual re-circulation cells) and not due to shock motion. All contour maps presented within this section were obtained by performing a tensor product linear interpolation of the results at the test points.

The buffet suppression envelope led to the following observations:

1.  SCBs positioned within $\pm 5\%$ of the mean shock location at the design point led to the suspension of shock oscillations sooner than those positioned further forward or further aft.
2.  Depending on the position of the SCB, the minimum threshold for buffet suppression existed at a height of $h_b/c = 0.002$.
3.  SCB profiles with relatively small crest heights ($0.001 \leq h_b/c \leq 0.005$) resulted in a buffeting flow-field, which could produce more excessive oscillations (particularly at placements more than $\pm 10\%$ away from the mean shock location).
4.  SCBs with crest heights $h_b/c \geq 0.007$ completely suppressed the buffet over the entire range of test positions.

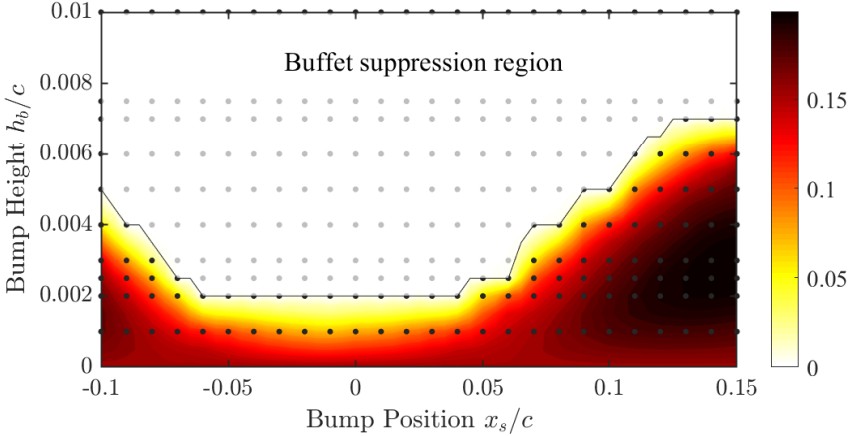

**Figure 16.** Peak-to-peak lift coefficient difference. ●: test point.

Observations (1) and (2) could be explained by the typical shock control rules introduced in the previous sensitivity study, as they were sufficiently tall enough to (a) provide an anchor for the front- and rear-shock legs of the $\lambda$-shock system and (b) result in the formation of the dual re-circulation system, which interfered with shock/separated shear-layer interaction. Observation (3) resulted as a consequence of the SCB geometry extending the "flat" region of the aerofoil curvature before (for positions aft of the mean shock location) or after (for positions fore of the mean shock location) the SCB tail, increasing shock travel. This effect was more pronounced in the case of SCBs that were

positioned closer to the trailing edge of the aerofoil, since the combination of the flow turning angle at the SCB tail combined with the inherent camber of the OAT15A promoted total flow reversal in the shear-layer and hence a more aggressive shock oscillation cycle compared to the baseline case. This also explained why reappearance of shock oscillation persisted at taller bumps further aft of the mean shock. The nature of observation (4) was a combination of the SCB having a tall enough crest to overcome the geometric deficiencies from Observation (3). The nature of control in this design space transitioned depending on the location of the SCB relative to the mean shock location. For SCBs aft of the mean shock location, the bump crest created an anchor point for the rear normal shock, which sat before the separated trailing edge shear-layer and exhibited a typical Type-B $\lambda$-shock structure. A strongly interacting dual re-circulation system existed within the trailing edge shear-layer, but could not interact with the normal shock due to the SCB tail angle. For the SCBs positioned forward of the mean shock location, the combination of bump ramp angle and leading edge curvature forced a splitting of the supersonic region, similar to the flow structure in Figure 12c. The dual re-circulation system presented here, however, compared to the aft SCB cases, exhibited a very weak interaction. The Mach number deceleration from the front-leg of the secondary $\lambda$-shock structure (which existed at the SCB crest position due to expansion) led to a very weak rear shock, which sat above the secondary re-circulation zone in a stable state.

Given that the SCB model in this study demonstrated that a substantial portion of the parametric space resulted in the complete quenching of shock motion, evaluation of the secondary objectives permitted the selection of an optimum SCB. As such, Figures 17–19 illustrate contour maps of the percentage difference in the mean lift and lift-to-drag ratio respectively with reference to the baseline mean performance. The region under which shock oscillations remained present were removed from the data; hence, evaluation of the aerodynamic performance was only considered for stable shock geometries.

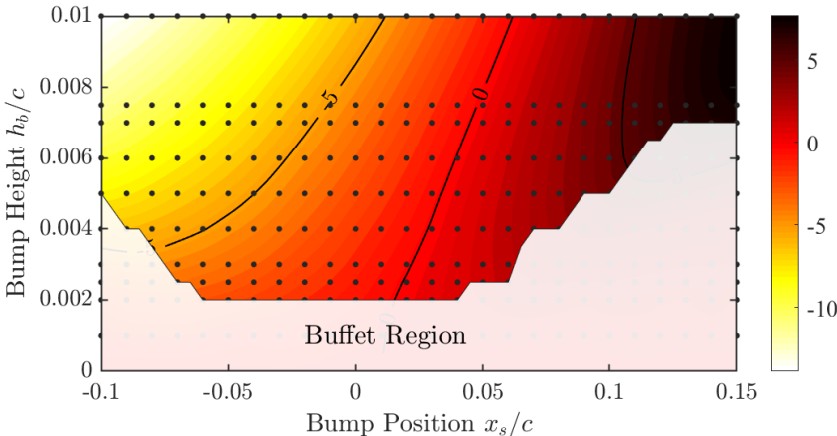

**Figure 17.** Percentage difference in the mean lift coefficient relative to the baseline. ●: test point.

The "0"-contour line on Figure 17 highlights the SCB geometries that yielded a recovery of the mean lift coefficient of the uncontrolled aerofoil in the test case. Lines at $\pm 5\%$ are also presented to help clarify how the lift coefficient varied with SCB position and height. The overall trends in lift coefficient variation strongly agreed with the combined observations of Figures 8a and 11a, whereby increasing the SCB crest height led to a depreciation in lift performance faster than the improved lift performance due to moving the SCB aft. Further, the lift performance tended towards a more linear variation with SCB position as height was increased. The geometries that lied on the "0"-mean lift contour line did not necessarily represent the "ideal" SCB in terms of pressure recovery that was typically sought after in stationary shock control devices, especially for taller SCBs, where the Mach re-acceleration peak became increasingly dominant. Following from the emergence of the different flow states that led to buffet suppression, the aerodynamic performance of forward positioned SCBs was severely

diminished due to the development of shock splitting and widening of the boundary layer from the interaction between the secondary re-circulation cell and terminating shock foot. Conversely, the SCBs with crest heights above $h_b/c = 0.0075$ provided the largest improvement in the overall relative lift coefficient with a $\geq 5\%$ increase achievable at $x_s/c \geq 0.10$. For these cases, the flow-field exhibited the same basic structure as explained under Observation (4); however, the difference was that the rear-leg of the $\lambda$-shock could sit up to 5% fore of the SCB crest position. This had the effect of extending the sonic flow region further along the aerofoil, increasing the width of the pressure rooftop, as well as providing a relatively small boundary layer thickness, despite there being fully reversed flow over the aft portion of the aerofoil.

There remained the existence of several SCB geometries within the parametric space that offered total suppression of shock oscillation and provided a resultant steady-state lift coefficient equal to or greater than the mean baseline. Given that these SCBs satisfied Criteria 1 and 2 of the design evaluation, Figures 18 and 19 identify that a large portion of the bump designs led to an overall improvement in the relative percentage variation in drag coefficient and lift-to-drag ratio, as there existed only a small window of forward sitting bump geometries that yielded a reduction and hence failed Criterion 3. The relative percentage difference in the drag coefficient suggested that the introduction of the SCB improved the overall drag performance of the aerofoil. This was seen most prominently in SCBs positioned close to the mean shock location, with a relatively large crest height. This effect likely developed from a combination of the comparatively small boundary layer thickness present aft of the SCB in spite of the poor shock characteristics for these configurations. It could further be implied that since these geometries led to a reduction in lift coefficient on the order of $\approx 5\%$; the low drag coefficient was indicative of a shift in the flight condition rather than a significant improvement in the performance. For this reason, the drag coefficient was not considered in isolation, but rather in the context of the lift-to-drag ratio. The most beneficial improvements to the lift-to-drag ratio were realised in the upper right quadrant of the test matrix, with SCBs that sat between $x_s/c = 0.05 : 0.15$ aft of the mean shock location and had a crest height within the $h_b/c = 0.0075 : 0.01$ range. The maximum relative percentage improvement to the aerofoil lift-drag ratio was achieved at $x_s/c = 0.11$ with a height of $h_b/c = 0.01$, which also offered a 5% increase in lift coefficient (shown in Figure 17). There was a small range of SCB geometries that, based on the parametric study performed, satisfied the design criteria. To understand the performance of these designs in off-design conditions, the area of apparent optimal configurations was analysed, which included the SCBs with heights of $h_b/c = 0.01$ that lied on the 0 and 5% line, as defined in Figure 17. For the remainder of this study, the SCB at position, $x_s/c = 0.06$, will be designated Design A and the SCB at position, $x_s/c = 0.11$, Design B. The mean pressure coefficients for these designs are presented in Figure 20 and the Mach number contours in Figure 21. In spite of yielding an increase in the lift-drag ratio, Design A still exhibited the adverse flow re-acceleration at the SCB crest; however, from Figure 21a, the impact on the overall supersonic region was smaller in comparison to the shock splitting that appeared for the SCB at 45%c (Figure 12c). This design case represented the minimum viable choice for off-design. Design B, conversely, offered the ideal pressure recovery, as the pressure recovery about the shock was improved considerably compared to the baseline aerofoil, and there was minimal flow re-acceleration compared to Design A. From Figure 21b, the resultant $\lambda$-shock structure resembled that of the optimum Type-B shape.

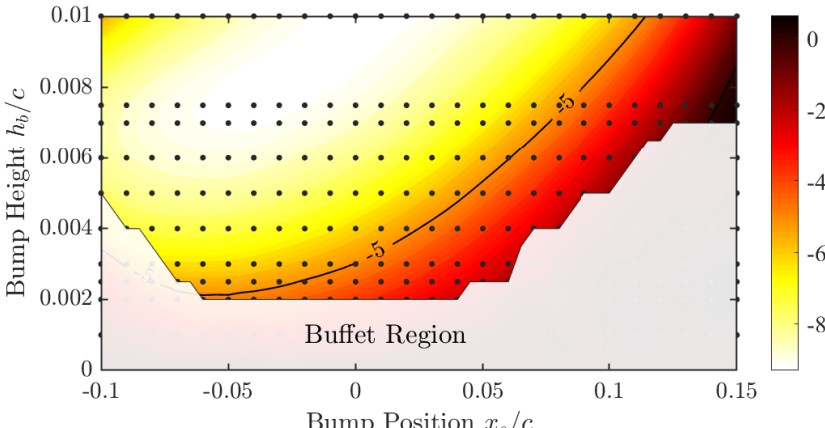

**Figure 18.** Percentage difference in the drag coefficient relative to the baseline mean. ●: test point.

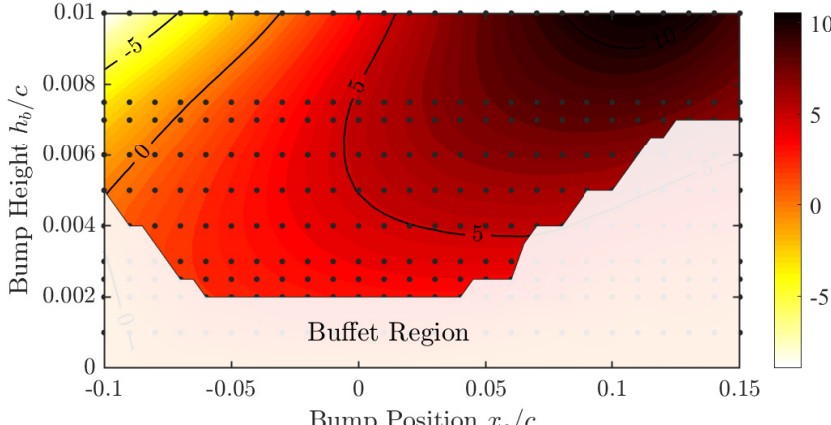

**Figure 19.** Percentage difference in the lift-to-drag ratio relative to the baseline mean. ●: test point.

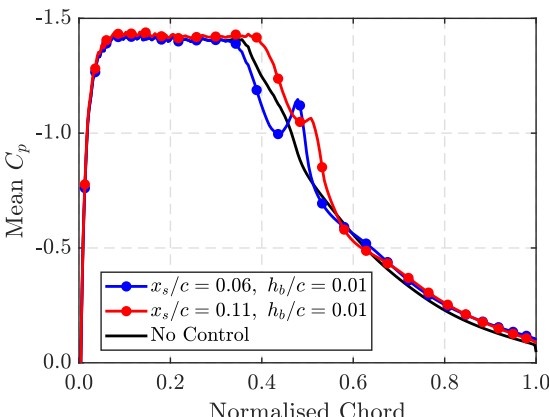

**Figure 20.** Mean pressure coefficient of SCB Designs A and B compared to no control.

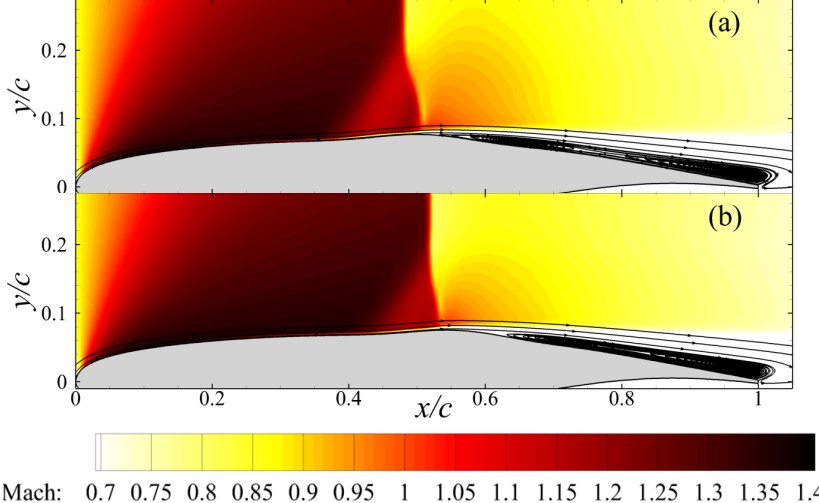

**Figure 21.** Mach contours and streamlines of the local flow-field for (**a**) Design A (**b**) Design B.

## 6. Off-Design Buffet Control

In order to assess the performance of the optimal SCB geometries in flight conditions outside of the design space, it was first necessary to evaluate the angle of attack range for which transonic shock oscillations developed, as well as to establish the lift profile of the aerofoil before the onset of the buffet. Given that Designs A and B were chosen for meeting or exceeding the selection criteria at the design flight condition, the off-design objectives for the SCB were as follows:

1.  Offset the buffet boundary to some angle greater than the design point, and
2.  Increase the maximum lift coefficient before stall/buffet onset, effectively extending the usable flight envelope

Figure 22 shows the mean and limit lift coefficient using the baseline computational URANS setup. Increments of $\Delta\alpha = 0.25°$ were made to the oncoming free-stream near the vicinity of buffet to capture onset and offset. The onset of the buffet was observed to appear at $\alpha = 3.25°$, at which point the mean lift coefficient continued to decrease and the fluctuations due to the presence of a shock oscillation increased. At angles greater than $\alpha = 5.0°$, the shock-wave boundary-layer interaction did not produce the stable, self-sustained oscillations as observed within the buffet envelope, and hence could be regarded as the buffet offset boundary.

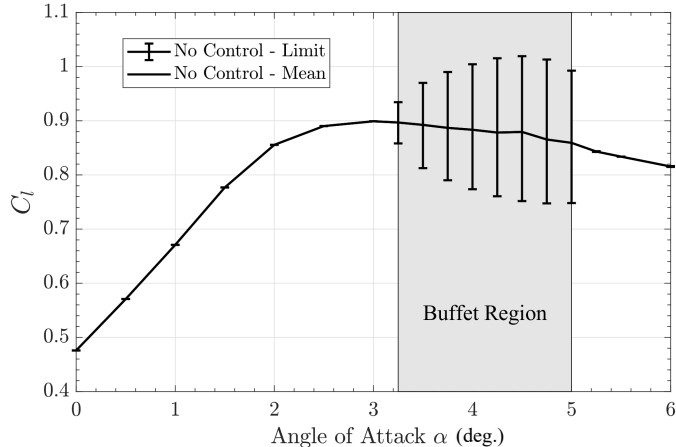

**Figure 22.** Mean and limit lift coefficient variation with the angle of attack at $M = 0.73$.

The mean pressure and RMS pressure coefficients are presented in Figure 23a,b, respectively, and served to illustrate the flow-field characteristics across the buffet envelope. These results showed that with increasing angle of attack from the onset, the mean shock location (identifiable by the peak in the RMS pressure curves) moved forward from $x_{sh}/c = 0.45 \rightarrow 0.34$, and the shock travel widened significantly at $\alpha = 4.5°$, then narrowing before offset. The peak intensity also increased over this transition, suggesting an increase in shock strength across the buffet cycle. This information was useful to consider under the on-design SCB framework established in the previous sections, as the position of the SCB was considered relative to the mean shock location. In the off-design context, this reference was no longer meaningful as the mean shock location also varied with respect to the angle of attack. As such, the definition of SCB position is represented as $x_{bc}/c$, which denotes the position of the bump crest relative to the leading edge as a percentage of aerofoil chord. Hence, Design A had position $x_{bc}/c = 0.51$, and Design B had position $x_{bc}/c = 0.56$.

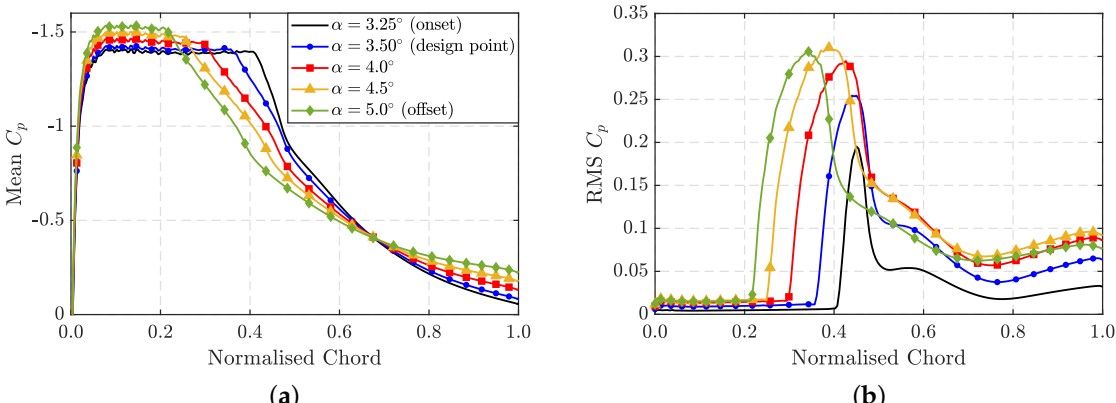

**Figure 23.** Pressure statistics variation with the angle of attack at $M = 0.73$. (**a**) Mean pressure coefficient. (**b**) RMS pressure coefficient.

The following simulations performed on Designs A and B were carried out by deforming the SCB at the onset of buffet using the same dynamic mesh motion used in the on-design case. All the following simulations were performed at the next flight condition using the already deformed mesh; thus, the transients that would appear during the transition phase between angles were omitted from this analysis, as it was beyond the scope of the study. Figure 24 shows the resultant mean and limit lift coefficients of Designs A and B over the buffet region with respect to uncontrolled aerofoil performance. From this result, it was revealed that both designs were capable of suppressing shock oscillations beyond their design points, though the performance of the two varied across the envelope. Design A, which was selected as the minimum viable design at the design flight condition, demonstrated the ability to offset buffet for the OAT15A entirely at $M = 0.73$. Given that Design A was optimised for the equivalent mean lift at $\alpha = 3.5°$, the SCB yielded a slightly lower lift coefficient at activation compared to the mean of the baseline; however, this decrease was only 2.4% below the mean value and, thus, did not severely affect the overall aerodynamic performance. This design also resulted in an increase of 6.4% of the maximum lift coefficient, which was obtained at $\alpha = 5.0°$. Design B provided an immediate improvement of the lift coefficient at onset and continued to extend the maximum lift coefficient by 7.6% at $\alpha = 4.5°$. However, at this point, the aerofoil with Design B deployed became susceptible to buffet instability. The peak difference in the lift coefficient was not sufficiently large to suggest a buffeting flow-field; however, there were small perturbations in the rear-shock-leg front that were typical of near-onset flow breakdown.

Further analysis of Designs A and B was deduced from the mean pressure coefficients, shown in Figure 25a,b. The trends that appeared in both of these results bore resemblance to the relationships determined in the analysis of SCB position relative to the mean shock location. For Design A in

particular, the evolution of the pressure spike at the bump gradually decreased, moving towards the "ideal" pressure recovery between $\alpha = 4.0° \rightarrow 4.5°$ before becoming almost indistinguishable at $\alpha = 5.0°$. A similar development was present in Design B; however, the transition occurred at an earlier angle of attack, after which the shock motion developed. This observation was not surprising, and its similarity with the SCB position sensitivity study could be obtained by examining the distance between the mean shock location at each angle of attack with respect to a fixed bump. As the angle was increased within the buffet envelope, the mean shock position moved further forward along the aerofoil (see Figure 23), which translated into the SCB designs being positioned increasingly aft of this reference point. Figure 26 illustrates the Mach contours at different angles within the SCB deployment space, for Design A, which further highlighted the interaction between the forward moving mean shock location with a fixed SCB. It was evident that the local flow on the aerofoil surface bore a strong correlation to the $\lambda$-shock structures characteristic of the variation in SCB position at the design point.

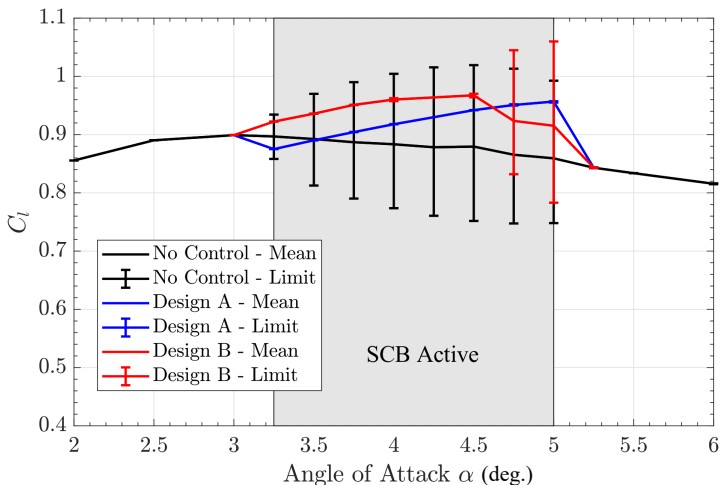

**Figure 24.** Comparison of the mean and limit lift coefficient variation with the angle of attack at $M = 0.73$ for Designs A and B.

From the tested designs in Figure 17, the lift coefficient varied roughly linearly between SCBs positioned $x_{cb}/c = 0.45 : 0.51$, and increases in lift were present up to $x_{cb}/c = 0.6$; however, due to sizing restrictions, this was the extent of the position range. Following that at $\alpha = 5.0°$, the mean shock location was at $x_{sh}/c = 0.34$, and the relative SCB crest positions of Designs A and B were $x_s/c = 0.17$ and $x_s/c = 0.22$. Design A did not result in shock buffet at this point. Conversely for Design B, the marker of a buffet instability occurred when the shock was at $x_s h/c = 0.39$, which had a relative crest position of $x_s/c = 0.17$. This suggested that buffet onset would likely occur for Design A if it was positioned any further aft of its current setting; moreover, this suggested that an extension of the buffet region (illustrated in Figure 16) for relative positions greater than $x_s/c = 0.17$. It is important to note that the efficacy of the SCB optimal geometries in the design point and off-design flight conditions appeared heavily linked to the inherent aerofoil curvature that existed independent of the control device. The flow structures that complimented shock oscillation suppression depended on the flatness of the aerofoil surface in front of the SCB position and the camber present about the trailing edge.

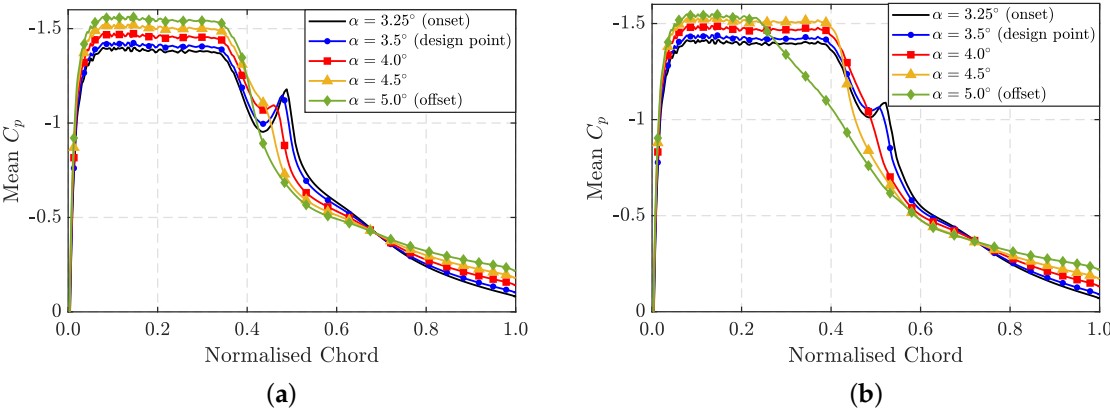

**Figure 25.** Mean pressure coefficients of Designs A and B at $M = 0.73$. (**a**) Design A. (**b**) Design B.

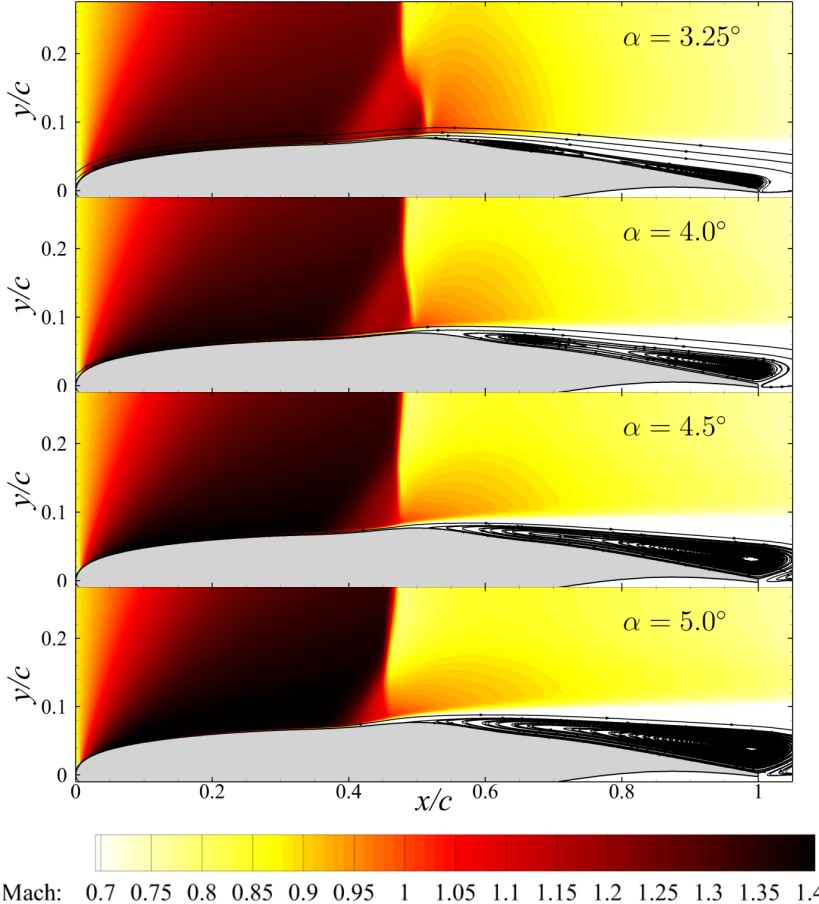

**Figure 26.** Mach contours and streamlines of the local flow-field of Design A at various angles within a clean buffet envelope.

## 7. Conclusions

The current research demonstrated the efficacy of contour-based shock control bumps to mitigate transonic shock oscillations and improve the aerodynamic performance of a 2D aerofoil section within the buffet envelope. The URANS model was used to validate the unsteady flow-field of the OAT15A with reference to experimental data and subsequently explore the design space through the activation of SCB geometries within a buffeting environment. At the design point $M = 0.73$, $\alpha = 3.5°$,

a parametric evaluation of the core geometric variables that defined the Hicks–Henne-derived SCB revealed an extensive parameter space for which complete suppression of shock wave oscillations existed. It was found that the deployment rate had no significant impact on the overall steady-state result, although the time taken to reach a stable shock system was increased for slower speeds. The SCB was found to control shock oscillations in general by: (a) weakening the strength of the normal shock present at the rear of the supersonic region by introducing a shock bifurcation structure known as a $\lambda$-shock; and (b) promoting the formation of a dual re-circulation cell system in the otherwise separated shear-layer, by augmenting the surface curvature. In the design condition, SCBs placed within $\pm5\%$ chord of the mean shock location resulted in shock control; however, this window extended as far as $-10 - 15\%c$ when the SCB crest height was above 0.75%c. In general, if the interaction between the dual re-circulation zones lead to a reformation of a singular cell, the buffet instability will develop, and often appearing with increased strength. For taller, and more aft SCB designs, however the physical boundary introduced by the SCB curvature combined with massively reduced rear shock strength, permits shock control in spite of the large singular re-circulation zone over the aft portion of the aerofoil. The parametric studies showed that there is a strong link between bump position and height, whereby moving the SCB crest position aft required a proportional increase in height to maintain the equivalent lift coefficient. This relationship however is also strongly dependent on the inherent curvature of the aerofoil upper surface, particularly in the aft portion of the OAT15A, where substantial camber is present. It was found that these types of SCBs, with crest positions between $51 - 60\%c$ and crest heights of $0.75 - 1.0\%c$, were capable of resulting in shock suppression, with a lift coefficient equal to or greater (up to 5%) than that of the mean baseline, while offering a $5 - 10\%$ increase in the lift-drag ratio. Of this design space, two SCBs were tested in off-design conditions, at a fixed Mach number, in order to evaluate the extent of shock oscillation control within the buffet envelope of the present aerofoil. It was found that for an SCB at 51%c and 1% crest height, shock oscillations were controlled within the observed buffet window, thus improving the maximum lift coefficient by 6.4% and extending the flight envelope by $2°$. By increasing the angles of attack, the mean shock location was shifted forward. This observation highlighted the symmetry between mean shock location and SCB crest position, such that the same performance trends were present as with moving the SCB position relative to a fixed mean shock. This research suggested that dynamically activated shock control bumps could provide an effective means of transonic shock oscillation control, as they could be deployed exclusively at the onset of buffet, where the SCB led to improved stability and aerodynamic performance, and stowed at flight conditions where it would present adverse effects.

**Author Contributions:** Conceptualization, J.A.G. and G.A.V.; methodology, J.A.G.; validation, J.A.G and N.F.G.; formal analysis, J.A.G.; investigation, J.A.G.; resources, G.A.V.; data curation, J.A.G.; writing, original draft preparation, J.A.G, N.F.G., and G.A.V.; supervision, G.A.V.; project administration, G.A.V. All authors read and agreed to the published version of the manuscript.

**Funding:** This research received no external funding.

**Acknowledgments:** The authors would like to thank Dr. Robert Carrese from RMIT University for his comprehensive insight into the transonic buffet phenomenon and for providing the preliminary test matrix from which the present work was developed. The computations in this paper were performed with the aid of Sydney University's High Performance Computing Cluster.

**Conflicts of Interest:** The authors declare no conflict of interest.

## Abbreviations

The following abbreviations are used in this manuscript:

| | |
|---|---|
| SCB | Shock Control Bump |
| SWBLI | Shock-Wave Boundary-Layer Interaction |
| TSB | Transonic Shock Buffet |
| (U)RANS | (Unsteady) Reynolds-Averaged Navier–Stokes |
| LCO | Limit Cycle Oscillation |

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
