# Peer review of "A Numerical Investigation of the Geometric Parametrisation of Shock Control Bumps for Transonic Shock Oscillation Control"

_fluids, doi:10.3390/fluids5020046_

Round 1

Reviewer 1 Report

Authors are numerically studying transonic flow control over an OAT15A aerofoil by means of unsteady RANS and geometry manipulation. Overall, the manuscript is well written and organized with a suitable background and introduction. In addition, authors show some interesting numerical outcomes and conclusions.

  • It would be important to evaluate the effect of the proposed shock control bump (SCB) on the drag coefficient, Cd, as well.
  • More importantly, how the skin friction coefficient (i.e., Cf vs. x/c) is affected by this local change in the aerofoil geometry?
  • Related with the previous question, is the flow separation zone (negative Cf values) significantly decreased by means of SCB?

Minor comments:

My suggestion is to carefully check the grammar and spelling in the entire manuscript.

Need to define acronyms when firstly introduced. There are several in the abstract.

In abstract: should it be ‘through’?

In abstract: ‘defines’?

Author Response

To Reviewer 1,

The authors are grateful for your comments and suggestions. Please see the attachment for a document containing our responses.

Regards,

Jack Geoghegan

Reviewer 2 Report

The paper is easy to read and the goals and outcomes of the research are well stated. The methodology is clearly explained and shows rigorous work. However, some consideration should be given to:

  • Additional emphasis on where the novelty emanates. Is it from the SCB type (or geometry) or from the methodology used to perform the design study?

  • The authors validated their CFD analysis against steady wind tunnel data and show good agreement. Nonetheless, the reviewer wonders if the lack of comparison against dynamic results is enough to be confident in the conclusion from the dynamic deployment study (4.1 Case 1, Impact of Deployment Frequency Variation)?

  • Some plots (Fig 8(a), Fig 10(a) or Fig 22) with the interval bars are not easy to interpret at first. It should be made clearer that they represent potential buffeting (as in Fig 20).

Author Response

To Reviewer 2,

The authors are grateful for your comments and suggestions. Please see the attachment for a document containing our responses.

Regards,

Jack Geoghegan
